# Cooperative Learning for Cost-Adaptive Inference

**Xingli Fang**
Computer Science
North Carolina State University

**Richard Bradford**
Collins Aerospace

**Jung-Eun Kim**
Computer Science
North Carolina State University

## Abstract

We propose a cooperative training framework for deep neural network architectures that enables the runtime network depths to change to satisfy dynamic computing resource requirements. In our framework, the number of layers participating in computation can be chosen dynamically to meet performance-cost trade-offs at inference runtime. Our method trains two Teammate nets and a Leader net, and two sets of Teammate sub-networks with various depths through knowledge distillation. The Teammate nets derive sub-networks and transfer knowledge to them, and to each other, while the Leader net guides Teammate nets to ensure accuracy. The approach trains the framework atomically at once instead of individually training various sizes of models; in a sense, the various-sized networks are all trained at once, in a "package deal." The proposed framework is not tied to any specific architecture but can incorporate any existing models/architectures, therefore it can maintain stable results and is insensitive to the size of a dataset's feature map. Compared with other related approaches, it provides comparable accuracy to its full network while various sizes of models are available.

## 1 Introduction

As deep neural network architectures continue to grow in size and complexity, it is increasingly costly to store and execute them. The trend is especially challenging in resource-constrained environments, such as embedded platforms or edge devices, where having a succinct model is a precondition for meeting time and space constraints. Hence, to enable systems to function in dynamically changing running environments and platforms under diverse resource allowances, we propose a novel training framework for adaptive real-time inferences. Note that, for the purpose of this work, the resource cost matters only in inference but not in training.

To achieve efficient models, several authors took the approach of re-designing the neural network architecture to have fewer depths or widths, or to have more computation-efficient kernels [15, 5, 12, 31, 11]. Another approach involved removing/deactivating insignificant neurons, as shown in [29, 40, 25]. Yet another alternative is knowledge distillation [8] which trains a smaller-size network by learning the distribution explicitly or implicitly from an original full-size network of high accuracy. As opposed to such static approaches, which individually customize a model for each specific task/platform or a certain inference path in advance, some other efforts provided various options to accommodate different platforms by scaling widths and depths as shown in [7, 4, 35]. Although the aforementioned approaches can trade off performance and resource with different model sizes, they must *pre*-define the requirements of computing resources in advance. Moreover, those architectures must be trained multiple times, specifically for each individual model size.

Unlike the existing work, our approach is able to provide an agile prediction at any inference cost on the fly; also, it is not tied to any specific architecture but can adopt any refined models. We propose a novel pipeline of knowledge distillation, namely, *Cooperative* training framework, which employs a cohort of three cooperating networks, two *Teammate nets* and *Leader net*. Teammate networks offer

Workshop on Advancing Neural Network Training at 37th Conference on Neural Information Processing Systems (WANT@NeurIPS 2023).

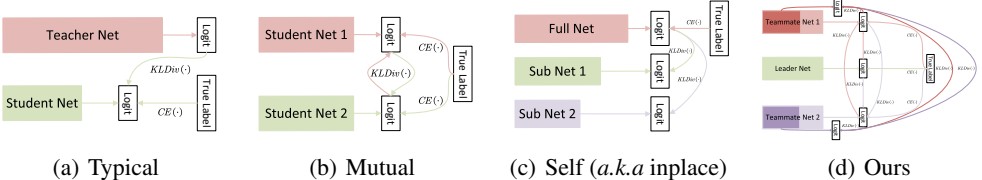

|   (a) Typical   |   (b) Mutual   |   (c) Self (*a.k.a* inplace)   |   (d) Ours   |

Figure 1: Overview of different knowledge distillations. (d) is enlarged in Fig. 2(b).

the soft labels to each other, and each of them derives sub-networks and transfers knowledge to the sub-networks. The Leader network is an auxiliary network learning from true labels and guiding Teammate nets to ensure accuracy. In particular, our approach trains the framework atomically at once instead of individually training various sizes of models; it does not simply train with labels and losses but also learns from numerous smaller networks at the same time. So, in a sense, the various-sized networks are all trained at once, in a "package deal," maintaining accuracy that is competitive with the original model.

## 2 Related Work

### 2.1 Static Approach to Compact Networks

To obtain a compact neural network, one approach is to re-design its architecture with fewer depths or widths, or with more computation-efficient kernels. Some manual network designs, such as SqueezeNet series [15, 5] and MobileNet series [12, 31, 11], have been applied to explore more efficient model architectures. Another way to reduce redundancy from existing models is to estimate the importance of neurons in a neural network and remove unimportant neurons. Some works explored how to estimate the importance, such as [29, 10], or deactivate unimportant neurons of a large network with sparse regularization [40] or greedy algorithm [25]. Another approach is knowledge distillation [8] which transfers the distribution of large networks to the light networks explicitly.

### 2.2 Scalable Architectures

As to scalable structures, prior works such as VGG series [32, 2], EfficientNet series [35, 34], MobileNet series [12, 31, 11] and others [30, 13] have proposed some architectures with scalable depth or width to meet the accuracy and static resource constraints. For instance, MobileNet series proposed three versions of MobileNet and adjusted the channel size of the main calculation cost part. One more example is ResNet [7] which adjusts the depth of the network by adjusting the number of repetitive convolution blocks at different stages divided by the down-sampling layers.

### 2.3 Adaptive Neural Networks

The whole idea of adaptive neural networks is to train a network to fit in a given environment and platform with various computing resources available to meet the dynamic constraints of real-world applications. An intuitive approach to training smaller networks of various sizes from a full-sized model is attaching intermediate (early) output branches on the forward pass, and training all the branches together as presented in [36] or [43]. [36] trained networks with multiple early exits. In [43], early exits imitate the main output's distribution to achieve better performance. However, such frameworks provide only a few choices, and the models' capacity or thresholds on the confidence need to be fixed in advance. More recent ideas are to re-design the structure of networks and optimize the training. [16] developed an adaptive deep neural network architecture that gradually adds layers to provide flexible options available concerning different timing constraints. [14] designed a stepped structure, MSD-Net, to produce time-adaptive classification results. [38] proposed RANet that offers an exiting branch at each stage and gradually adds more layers. In other studies, the width of networks is explored. [42, 22] tried training adaptive width networks by knowledge distillation. [41] proposed US-Net for selecting any channel width of convolution networks trained with self-distillation. Similarly, [9] trained a Dynamic BERT with dynamic depths and widths. Based on US-Net, [39] proposed MutualNet and pointed out that multi-scale inputs can help an adaptive model

be trained better. [20] proposed a CGM module making a decision on pruning a layer by considering the outputs of preceding layers and required scales. Li et al. [21] built DS-Net, including an online network and target sub-network, with a dynamic gating to choose a proper channel pruning rate.

## 2.4 Knowledge Distillation

The lottery ticket hypothesis [3] suggests that the existing refined neural networks have the potential to retain comparable accuracy even after removing or deactivating some nodes. One well-known way to achieve this is knowledge distillation. The original approach to knowledge distillation is to train a smaller-size neural network against a full-size neural network's outputs so that the small-size network can perform competitively. Moreover, [43] introduced a knowledge distillation method to train parameter-shared networks by creating multiple branches for different costs of inference. [42] applied a similar training policy at the channel level and switchable batch normalization to solve the problem of different means and variances of the aggregated feature led by ever-changing input channel size in preceding layers. [41] extended this adaptability from only several fixed ratios to any channel pruning rate according to the sandwich rule (see Sec. 4.3). [44] let two models from the same architecture learn each other's distribution of outputs to enhance the performance of both models. [28] introduced a teaching assistant network (an intermediate-size homogeneous network) between the teacher and student network to help the student network achieve more stable and better performance. [6] proposed an ensemble mutual learning and explored how to produce a good ensemble output.

## 3 Background

### 3.1 Knowledge Distillation

In a typical knowledge distillation [8] as in Fig. 1(a), there is a teacher and a student network. A teacher net is trained first, and then its outputs and accurate labels "teach" (*i.e.*, train) a student net. The training process for the student network is described as,

$$\mathcal{L}_s = (1 - \lambda)\mathcal{L}_{ce} + \lambda\mathcal{L}_{kl} \tag{1}$$

where $\mathcal{L}_{ce}$ is a cross-entropy loss, $\mathcal{L}_{kl}$ is Kullback-Leibler divergence loss [18], and $\lambda$ is to balance two losses. Before being fed into a softmax activation function of the teacher and student net, let each last input be $a_t$ and $a_s$, respectively. Then, the output of teacher net is $y_t = \mathsf{softmax}(a_t/\tau)$, where $\tau$, as the temperature, is to adjust the smoothness of probability distribution, and that of student net is $y_s = \mathsf{softmax}(a_s/\tau)$. Thus, $\mathcal{L}_{kl}$ is represented as,

$$\mathcal{L}_{kl} = \tau^2 KLDiv(y_s, y_t) \tag{2}$$

where $KLDiv(\cdot, \cdot)$ is Kullback-Leibler divergence loss function and $\tau$ is typically greater than or equal to 1. $\mathcal{L}_{ce}$ is formulated as,

$$\mathcal{L}_{ce} = CE(y_s, y_{true}) \tag{3}$$

where $CE(\cdot, \cdot)$ is a cross entropy loss function and $y_{true}$ is a true label from supervised dataset.

### 3.2 Mutual Knowledge Distillation

With an observation that logits of one model can help another model improve its performance, *mutual knowledge distillation* (deep mutual learning) was proposed by [44]. As depicted in Fig. 1(b), two models can learn from each other using formula (1), and significant improvement has been shown in each model's performance.

### 3.3 Self-Distillation

In self-distillation (inplace distillation), the teacher net becomes the student net itself, in which a student net is a subset of the full net (Fig. 1(c).) There is no pre-trained reference net since there is only a single net to be trained. According to [42, 41, 43, 39, 21], a common loss calculation is,

$$\mathcal{L}_{sl} = CE(y_{full}, y_{true}) + \lambda \sum_{i=1}^{n} \tau^2 KLDiv(y_{sub,i}, y_{full}) \tag{4}$$

where $y_{full}$ is the output from a full network, $y_{sub,i}$ is the output from $i$-th sub-network, $n$ is the total number of sub-networks designed manually prior to training, and $\lambda$ is a hyper-parameter to balance the two types of losses.

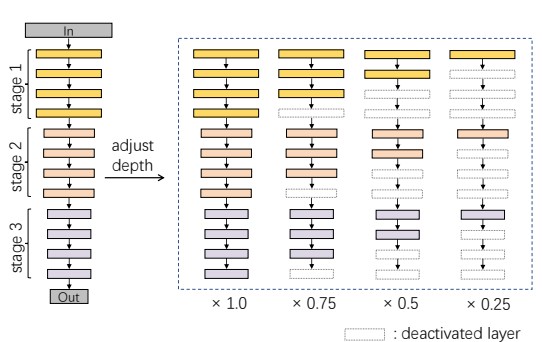 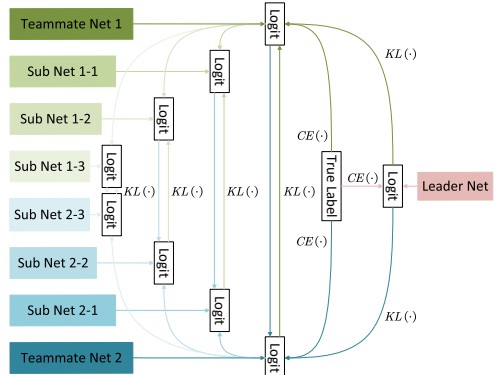

(a) Deriving sub-networks.        (b) The entire Cooperative training framework.

Figure 2: Overview of our approach.

# 4 Multi-Model Cooperative Learning

## 4.1 Deriving Sub-networks

Deep neural networks usually contain multiple stages composed of layers (blocks.) In general, a network's depth is a factor impacting performance and resource cost trade-offs. Hence we derive various-sized sub-nets composed of a subset of layers from the original net as in Fig. 2(a). Then, the knowledge of the original net is transferred to the sub-nets, *i.e.*, self-learning. A deeper sub-net can be constructed by adding more layers on top of a shallower sub-net. The number of layers employed in a sub-net calculates a *scaling factor*: *e.g.,* if a sub-net is formed by a quarter of layers in the original, the scaling factor of the sub-net is $0.25$. The largest sub-net is always the same size as the original net (*i.e.,* scaling factor $= 1.0$.) Depending on the number of stages and layers, the possible number of intermediate nets, granularity of scaling factor, etc. can vary. With given input image, $x$, scaling factor, $s$, and the corresponding output, $y_{apt}(\cdot, \cdot)$, the pass-through of our Cost-adaptive network is formulated as,

$$y_{apt}(x, s) = \mathsf{OUT}(\bigcirc_{j=1}^{u} \bigcirc_{i=1}^{sr^{(j)}} F^{(j,i)}(\mathsf{IN}(x))) \tag{5}$$

where the large $\bigcirc$ represents repeated function compositions, $\mathsf{OUT}(\cdot)$ is the function of the final layer, and $\mathsf{IN}(\cdot)$ is the function of the first layer. Also, $u$ is the number of stages in the network, $r^{(j)}$ is the number of repetitive blocks at the $j$-th stage, and $F^{(j,i)}(\cdot)$ is the $i$-th block at the $j$-th stage.

## 4.2 Flexible Depth Control: Masking Approach

To activate/deactivate each layer of a network for a flexible size, a binary mask is used. A common masking method in training is the Gumbel-Max trick [27]. In inference, the execution of an insignificant layer is deactivated by the mask on the fly. The binary gating mask $B^{(i)}$ is 1 if the $i^{th}$ layer is activated, otherwise 0. However, there are two challenges obstructing the employment of masking. One is that the masking schemes presented in the previous work [23, 37] may make the network sparser, but may not reduce non-parallel computations. The other challenge is that a binary mask cannot directly back-propagate since it is not differentiable. For the first challenge, we design a global mask to determine which layers would be better skipped. Given a residual neural network, if the structural differences are ignored, the $i$-th layer's forward propagation can be described as follows:

$$y^{(i)}(x) = F^{(i)}(x) + x \tag{6}$$

where $F^{(i)}(\cdot)$ represents the $i$-th layer calculating the feature map, $y^{(i)}(\cdot)$. Once the mask is applied, the $i$-th layer's forward propagation in training is represented as,

$$y_{m-train}^{(i)}(x) = F^{(i)}(x) \odot B^{(i)} + x \tag{7}$$

where $\odot$ denotes a broadcast multiplication. For an evaluation stage, the execution of the layers can be determined by the masking value $B$ as shown in (8):

$$y_{m-eval}^{(i)}(x) = \begin{cases} F^{(i)}(x) + x, & \text{if } B^{(i)} = 1 \\ x, & \text{if } B^{(i)} = 0 \end{cases} \tag{8}$$

That is, in an evaluation stage, the mask becomes a signal telling whether a layer is executed or not. The global mask, $B$, over all layers can be formulated as follows:

$$B = \arg \underset{i}{\mathsf{Top}} k(\log(\pi_i) + g_i) \tag{9}$$

where $g_i$ is noise sampled from Gumbel distribution and $\pi_i$ is a probability calculated by the mask. The mask values with the top-k scores are set to 1, and the others to 0. However, the mask is not differentiable yet, so the softmax is applied to create the following differentiable approximation,

$$\phi^{(i)}(p) = \frac{\exp((\log(\pi_i(p)) + g_i(p))/\tau)}{\sum_{j=1}^{n} \exp((\log(\pi_j(p)) + g_j(p))/\tau)} \tag{10}$$

where $g_i(\cdot)$ denotes the noise sampled from a standard Gumbel distribution, to randomize the sampling process. $\tau$ can be dynamically changed to control the smoothness of the outputs' distribution. It ranges from 0 to 1 (excluding 0.) When close to 0, the mask's output is approximately regarded as a binary value. $n$ is the number of total layers. In our evaluation, we set $\tau = 2/3$ as presented in [26, 23]. $\phi$ can match $\arg \max$ precisely so that it can make a mask gradually change its value from the continuous value to a value close to a binary during training. However, it cannot match the $\arg \mathsf{Top} k$ in the same way since $\phi$ is almost one hot value when $\tau$ approaches 0. Hence, we apply the Gumbel-Softmax re-parameterization trick to achieve a back-propagation with binary values as,

$$B = \mathsf{stopgrad}(\arg \underset{i}{\mathsf{Top}} k(\phi^{(i)}) - \phi) + \phi \tag{11}$$

where the $\mathsf{stopgrad}(\cdot)$ is a stop-gradient operation. As a result, by (11), the mask is both binary and differentiable.

## 4.3 Scaling Factor Sensitive Loss SFSL

We empirically recognized that accuracies of sub-nets are bounded by the shallowest and deepest net: a similar trend, but in width-controlled (channel-wise) sub-nets, is also addressed by Yu and Huang [41] which is called the sandwich rule. On top of that, we have observed that the layers composing the shallower net have a higher impact on the overall performance since the layers are "commonly" contained in all the other (sub-)nets. With this observation, we introduce a loss function, *Scaling Factor Sensitive Loss* (SFSL), to explicitly add more weights to the layers of shallower sub-nets.

For loss functions in knowledge distillation, one important source is Kullback-Leibler divergence loss between student, $y_s$, and teacher, $y_t$, as in (2). Another is cross-entropy loss between teacher, $y_t$, and true label, $y_{true}$, or between student, $y_s$, and true label, $y_{true}$, as in (3). As the model contains multiple ensembles, all losses are aggregated as in (4). More importantly, we weigh a shallower network by dividing the loss by its scaling factor which returns a larger value. The loss function of a sub-network, $\mathcal{L}_{subs}$, is as follows:

$$\mathcal{L}_{subs} = \sum_{i=1}^{n} \frac{\tau^2}{s} KLDiv(y_{sub,i}, y_{full}) \tag{12}$$

where $s$ is a scaling factor of depth (*i.e.*, size) of a sub-network over the full network. Then, the total loss function of the entire network, $\mathcal{L}'_{sl}$, is,

$$\mathcal{L}'_{sl} = CE(y_{full}, y_{true}) + \lambda \mathcal{L}_{subs} = CE(y_{full}, y_{true}) + \lambda \sum_{i=1}^{n} \frac{\tau^2}{s} KLDiv(y_{sub,i}, y_{full}) \tag{13}$$

The evaluation result is also aligned with our preconception that the proposed loss function can help shallower sub-networks achieve better accuracy than just uniformly accumulating the total loss.

## 4.4 Cooperative Training Framework

We introduce Self-, Interactive, and Guided learning, forming Cooperative training framework, enabled by two Teammate networks and their subnetworks and a Leader network.

**Self-Learning** In Self-learning, the sub-networks distill the knowledge from a full net. Based on (4), the full net is trained by cross-entropy loss and other sub-nets are trained by Kullback-Leibler divergence loss using soft labels produced by the full net. SFSL is applied with $\tau = 1$ and $\lambda = 1$. The loss function of self-learning is as follows:

$$\mathcal{L}_{apt\_sl} = CE(y_{apt}(x, 1.0), y_{true}) + \sum_{s_i \in \mathcal{S}} \frac{1}{s_i} KLDiv(y_{apt}(x, s_i), y_{apt}(x, 1.0)) \tag{14}$$

where $\mathcal{S}$ is a collection of scaling factors of the all sub-networks except for $s_i = 1.0$.

**Interactive Learning**   With only self-learning, there is no source to produce soft labels the full network can learn, possibly leading to imbalanced performances between the sub-networks. For the issue, we employ two *Teammate networks* offering soft labels for each other. A teammate net, as a full network, is tagged along with a cohort of self-learned sub-nets as shown in Fig. 2(b). It is called *interactive learning* since the two are learning from each other. The two Teammate nets may or may not be from the same architecture. Let $y_{apt,a}$ and $y_{apt,b}$ be the outputs produced by the two Teammate nets. The same scaling factor is chosen to generate their sub-nets. Then, the loss of interactive learning for the target network can be represented as:

$$\mathcal{L}_{apt\_cl,a} = \sum_{s_i \in \mathcal{S}'} KLDiv(y_{apt,a}(x, s_i), y_{apt,b}(x, s_i)) \tag{15}$$

where $\mathcal{S}'$ is a set of scaling factors including $s_i = 1.0$. As one Teammate network takes a turn to learn from the other, we just swap the "teacher" and the "student" for the arguments of Kullback-Leibler divergence loss calculation in (15) as,

$$\mathcal{L}_{apt\_cl,b} = \sum_{s_i \in \mathcal{S}'} KLDiv(y_{apt,b}(x, s_i), y_{apt,a}(x, s_i)) \tag{16}$$

On top of that, since two networks that learn from each other can also improve each other's performance as shown [44], the combined loss is calculated as follows:

$$\mathcal{L}_{apt\_cl} = \mathcal{L}_{apt\_cl,a} + \mathcal{L}_{apt\_cl,b} \tag{17}$$

**Guided Learning**   Even with self-learning and interactive learning, we observed that a balance of losses may still not be preserved. Hence, we introduce *Leader Network* which only learns the knowledge from the true labels. As a result, a full-size sub-net in the cohort of a Teammate net can learn the distribution from this highly accurate net as follows:

$$\mathcal{L}_{apt\_ol} = CE(y_{leader}, y_{true}) + \sum_{i \in \{a,b\}} KLDiv(y_{apt,i}(x, 1.0), y_{leader}) \tag{18}$$

where $y_{leader}$ denotes the output by the Leader net, and $i$ denotes which Teammate net is used for calculation.

**Total loss**   Finally, all three losses are combined together. The total loss is formulated as follows:

$$\mathcal{L}_{apt\_total} = \sum_{i \in \{a,b\}} \mathcal{L}_{apt\_sl,i} + \mathcal{L}_{apt\_cl} + \mathcal{L}_{apt\_ol} \tag{19}$$

where $\mathcal{L}_{apt\_sl,i}$ is Teammate network $i$'s self-learning loss. Note that we stop the gradients of the outputs produced by networks when we use them as labels.

## 5   Experiments

### 5.1   Experimental Setting

#### 5.1.1   Datasets

**CIFAR-100 [17]:**   the images are zero-padded in 4 directions, and $32 \times 32$ crops are randomly sampled or their horizontal flips normalized with the per-channel mean and standard deviation. Due to the smaller image size than ImageNet, the first downsampling in ResNet is replaced by a convolution with $3 \times 3$ kernel and a stride of 2 before the global average pooling.

**Tiny ImageNet [19]:**   Tiny ImageNet is derived from ImageNet [1]. Since labels of testing images are not publicly available, the validation images are used for evaluation. To make the number of input channels the same among all samples, the gray-scale images are converted into RGB images. For training, we horizontally flip the images randomly for data augmentation while keeping the original format of validation images.

#### 5.1.2   Network architectures for comparisons

**Baseline**   we use a ResNet152 [7] trained only with cross-entropy loss as a baseline. For comparisons, the number of layers in each stage is adjusted to the same proportion as the sub-network that we generate.

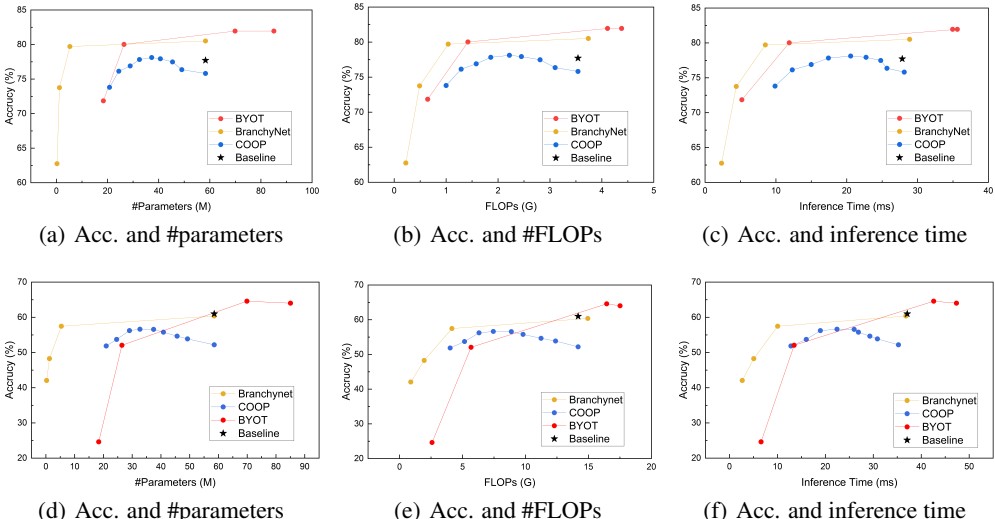

(a) Acc. and #parameters     (b) Acc. and #FLOPs     (c) Acc. and inference time

(d) Acc. and #parameters     (e) Acc. and #FLOPs     (f) Acc. and inference time

Figure 3: Performance and costs of different methods on CIFAR100 (top row) and Tiny ImageNet (bottom row.) (Only the full size of `Baseline` is shown in the charts since the subnets' accuracies are incomparable to the others.)

**BranchyNet**    BranchyNet [36] creates muti-branch networks by adding additional exits. While ResNet110 was adapted in the original, we adapted and fine-tuned ResNet152 by adding one more stage. We implemented 3 branches for each stage. Each 1st, 2nd, and 3rd branch includes 3, 2, and 1 convolutional blocks, respectively. A global average pooling layer and a fully-connected layer follow the convolutional blocks in each branch. BranchyNet takes a weighted sum of the cross-entropy losses over all exits. Since precise settings for weights are not presented in the original paper, we set the weights as $\{0.4, 0.6, 0.8, 1.0\}$ from the first exit to the final.

**BYOT**    'Be your own teacher' method [43] is also designed to achieve a multi-cost inference by employing additional exits in the early stages. Its loss function and training method make it different from BranchyNet. In addition to training each exit, `BYOT` also applies the knowledge distillation in part and Euclidean distance to train the exits with the knowledge of the final exit to better train the outputs of the exits.

### 5.1.3 Hyper-parameters

For learning rates and epochs: we train a model for 200 epochs using learning rates of $\{1 \times 10^{-1}, 1 \times 10^{-2}, 1 \times 10^{-3}, 1 \times 10^{-4}\}$ - the 75th, 130th and 180th epochs are changing points of learning rates. The learning rate of the first epoch is set to $1 \times 10^{-2}$ to warm up the network. For optimizer, the optimizer we adapt is SGD. We set the momentum to 0.9 and weights decay to $5 \times 10^{-4}$. We report results with the model at the final training epoch.

### 5.1.4 Platforms and running environments

The models are trained on the GPU cluster server running on Ubuntu with various GPU resources including NVIDIA RTX A6000, NVIDIA RTX 6000, NVIDIA RTX 5000, NVIDIA GeForce GTX 1080 Ti, and NVIDIA GeForce GTX 750 Ti. For evaluation, we run models on python 3.7 on Windows 11, and use NVIDIA GeForce RTX 2060 to measure performance metrics. For measuring the number of parameters and FLOPs, we use the python package *THOP* to count them. The inference time is evaluated with the mini-batch size of 1 input which is the same as the data size in the datasets on NVIDIA GeForce RTX 2060. We run 100 times to warm up the GPU before we start any evaluation, and compute the inference time using the average of 1,000 times' results. All models are implemented and trained in Pytorch.

Table 1: Networks' performance on CIFAR-100 and TinyImageNet. ('Exe.' stands for execution, 'Param.' stands for parameters, 'Inf.' stands for inference time, and 'Acc.' stands for accuracy.)

| Network | Exe. Portion | CIFAR-100 | | | | TinyImageNet | | | |
|---|---|---|---|---|---|---|---|---|---|
| | | #Param. | #FLOPs | Inf. (ms) | Acc.(%) | #Param. | #FLOPs | Inf. (ms) | Acc.(%) |
| Baseline | $0.2\times$ | 20.78M | 1.00G | 9.55 | 7.65 | 20.98M | 4.02G | 11.70 | 6.07 |
| | $0.4\times$ | 28.88M | 1.58G | 15.09 | 10.63 | 29.08M | 6.31G | 18.09 | 9.50 |
| | $0.6\times$ | 37.26M | 2.22G | 19.84 | 26.04 | 37.46M | 8.88G | 24.30 | 15.01 |
| | $0.8\times$ | 45.43M | 2.81G | 23.62 | 47.84 | 45.63M | 11.24G | 30.06 | 32.57 |
| | $1.0\times$ | 58.34M | 3.54G | 27.84 | 77.71 | 58.55M | 14.18G | 35.29 | 60.98 |
| BranchyNet | $1^{st}$ exit | 243.36K | 223.18M | 2.33 | 62.75 | 243.36K | 892.65M | 2.67 | 42.05 |
| | $2^{nd}$ exit | 1.21M | 488.95M | 4.41 | 73.76 | 1.21M | 1.96G | 5.07 | 48.25 |
| | $3^{rd}$ exit | 5.29M | 1.04G | 8.51 | 79.71 | 5.29M | 4.16G | 10.05 | 57.46 |
| | $4^{th}$ exit | 58.34M | 3.74G | 28.88 | 80.51 | 58.55M | 14.95G | 36.81 | 60.35 |
| BYOT | $1^{st}$ exit | 18.39M | 644.25M | 5.20 | 71.85 | 18.39M | 2.58G | 6.60 | 24.62 |
| | $2^{nd}$ exit | 26.45M | 1.42G | 11.89 | 80.02 | 26.45M | 5.68G | 13.49 | 52.05 |
| | $3^{rd}$ exit | 69.90M | 4.11G | 35.62 | 81.95 | 69.90M | 16.45G | 42.60 | 64.57 |
| | $4^{th}$ exit | 85.07M | 4.38G | 34.95 | 81.95 | 85.07M | 17.51G | 47.35 | 64.01 |
| COOP | $0.2\times$ | 20.78M | 1.00G | 9.88 | 73.80 | 20.98M | 4.02G | 11.39 | 51.86 |
| | $0.4\times$ | 28.88M | 1.58G | 14.99 | 76.89 | 29.08M | 6.31G | 18.02 | 56.22 |
| | $0.6\times$ | 37.26M | 2.22G | 20.55 | 78.11 | 37.46M | 8.88G | 24.53 | 56.56 |
| | $0.8\times$ | 45.43M | 2.81G | 24.80 | 77.47 | 45.63M | 11.24G | 29.29 | 54.65 |
| | $1.0\times$ | 58.34M | 3.54G | 28.12 | 75.81 | 58.55M | 14.18G | 35.04 | 52.19 |

## 5.2 Experiments on CIFAR-100

In this section, we present the experimental results on CIFAR-100. Both `BranchyNet` and `BYOT` have four exits including the original exit. We display accuracy according to # FLOPs, # parameters and inference time in Fig. 3 (top row) and Table 1 left half. Some result points show that `BranchyNet` and `BYOT` perform better than our approach, `COOP`, in terms of accuracy. That is because these two methods yield hierarchies of features for classifier, which is empirically shown by a series of works such as FCN [24] - that is, multi-scale feature maps' fusion is beneficial to the whole network's performance. Another factor, which is supported by GoogLeNet [33], is that early exits can improve model training to some extent. However, for inference, multi-scale outputs may show unpredictable and unstable performance drops depending on the feature maps' sizes. We will discuss this later in Sec. 5.3. Our approach, `COOP`, achieves accuracy close to `Baseline`. Although its performance does not outperform `BranchyNet` and `BYOT` at all measurement points, there are three strengths of `COOP`: i) It provides more flexible options for computing capacity, ii) the size of the output feature map is consistent with the size of the original network so that our method is an architecture-free (general) method, and iii) it provides a predictable and smoother decrease in accuracy as the model's depth decreases.

## 5.3 Experiments on Tiny Imagenet

In this section, we present the experimental results on Tiny ImageNet in Fig. 3 (bottom row) and Table 1 right half. Since the size of feature maps is larger than that in CIFAR-100, we can examine the impact of feature maps' size by comparing the results on the two different datasets. One thing to note in the right half Table 1 is that `BYOT` shows significantly poorer performance on $1^{st}$ and $2^{nd}$ exits compared with the results on CIFAR-100. This is because the feature map in Tiny ImageNet is $64 \times 64$, which is four times as large as that in CIFAR-100. Hence, the much smaller feature map size greatly degrades the performance of `BYOT`. On the other hand, `COOP` and `BranchyNet` show a more stable trend, since all branches are independent in the training phase. However, `BranchyNet`'s loss function contains hyper-parameters that need to be set manually and fine-tuned, and the exits and loss functions have to be carefully designed case by case. In contrast, `COOP` is more convenient since it does not need to be "re-designed" when other existing refined networks are applied. Moreover, although the accuracy of the model may not be the best in all cases, it shows the highest accuracy in most of the middle range of scaling factors. In light of the motivation of the work which would utilize compact and efficient models, the trend shows the kind of utility we are seeking.Overall, `COOP` generally shows less degraded performance when the depth of the model decreases.

## 5.4 Ablation Study

We explore and show the impacts of the components contributing to the network's performance with CIFAR-100.

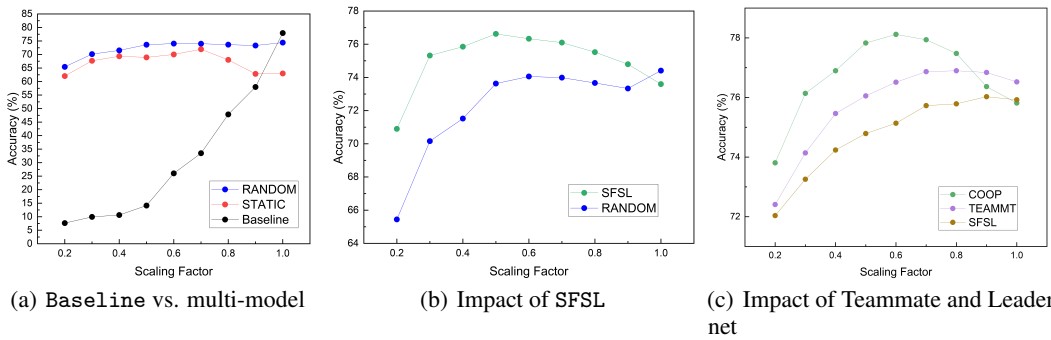

| (a) `Baseline` vs. multi-model | (b) Impact of SFSL | (c) Impact of Teammate and Leader net |

Figure 4: Ablation experiments

### 5.4.1 Impact of Multi-Model Training

For multi-model training, we employed 4 sub-networks to train. Scaling factors of the sub-nets are sampled in two ways:

- `STATIC`: static values are set, $[1.0, 0.7, 0.4, 0.2]$.
- `RANDOM`: other than the deepest $(1.0\times)$ and shallowest net $(0.2\times)$, two scaling factors are randomly sampled from $[0.3, 0.9]$ with a step of $0.1$ in every training epoch.

The approaches' comparison result with `Baseline` is shown in Fig. 4(a). Considering `Baseline` is merely trained by cross-entropy loss, the result shows that the multi-model training strategies can significantly help sub-networks' performance in most cases. In addition, `RANDOM` outperforms `STATIC`, which demonstrates how the random depths sampling provides stable performance in any size of the model by training various sub-networks. Accordingly, in the following results in the section,

Table 2: Performance comparison (accuracy) of different methods for adaptive inference.

| Network | Member | Acc.(%) | | | | |
| | | 0.2× | 0.4× | 0.6× | 0.8× | 1.0× |
|---|---|---|---|---|---|---|
| SFSL | - | 70.89 | 75.85 | 76.33 | 75.52 | 73.59 |
| TEAMMT | Teammate | 72.40 | 75.46 | 76.51 | 76.89 | **76.52** |
| COOP | Leader | n/a | n/a | n/a | n/a | 79.04 |
| | Teammate | **73.80** | **76.89** | **78.11** | **77.47** | 75.81 |

we employ random sampling for scaling factors. Fig. 4(b) shows the results of training with SFSL on top of `RANDOM`, which we call SFSL. It clearly shows that SFSL enhances the overall performance. It is noticeable that even the smaller sub-networks show very much enhanced performance by SFSL, which explains the impact of weighting smaller networks' loss more in the overall loss.

**Impact of Teammate and Leader networks**   We show the impact of Teammate and Leader nets with the following approaches,

- `TEAMMT`: on top of SFSL, one more Teammate network and a cohort of its subnetworks are added - two Teammate nets in total. This corresponds to *Interactive Learning* introduced in Sec. 4.4.
- `COOP`: the proposed, *Cooperative learning* framework, that is, Leader net is added on top of `TEAMMT`.

The results are in Table 2 and Fig. 4(c). A general trend is that adding Teammate nets enhances performance (SFSL vs. `TEAMMT`), and adding Leader net also does (`TEAMMT` vs. `COOP`.) In particular, Interactive learning (`TEAMMT`) helps the network achieve better performance in all scaling factors. Cooperative learning (`COOP`) even boosts the performance further in a wide range of factors.

## 6   Conclusion

In this paper, we demonstrate that our approach (Cooperative training framework) dynamically resizes networks to meet resource constraints while offering more size options, applying seamlessly to various neural network architectures with minimal additional inference cost. Our approach efficiently trains multiple different-sized subnetworks simultaneously, as evidenced by experiments where even smaller-sized networks perform comparably to the full-sized network.

## Acknowledgments and Disclosure of Funding

This publication is supported by the National Science Foundation under Grant No. 1945541 (transferred and extended to No. 2302610). Any opinions, findings, and conclusions or recommendations expressed in this material are those of the author(s) and do not necessarily reflect the views of the National Science Foundation.

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
