# OpenReview forum: "Cooperative Learning for Cost-Adaptive Inference"
_NeurIPS.cc/2023/Workshop/WANT — WANT@NeurIPS 2023 Poster_

### Official Review · Reviewer_UwsV · 2023-10-22
**Training Adaptive and Efficient Neural Networks via Cooperative Learning**

**Confidence:** 3

**Review:**

The paper addresses the problem of enabling neural networks to adaptively change their size to meet different resource constraints at inference time, important for deploying models efficiently in real-world systems.

The proposed cooperative training framework with self-learning, interactive learning, and guided learning appears novel. Training sub-networks of different depths together and mutually distilling knowledge between them is interesting.

The method seems architecture-agnostic, which is useful since it could be applied to enhance different pre-trained models without much modification.

Results on CIFAR and Tiny ImageNet datasets demonstrate the approach can lead to good performance of sub-networks comparable to the full network, with smooth and predictable accuracy drops as size decreases.

Issues:

1. Accuracy of sub-networks does not clearly outperform other approaches like BranchyNet and BYOT in the experiments.
2. More analysis could be provided to understand the contributions of the different components (self-learning, interactive learning, guided learning).
3. More measurements on actual mobile platforms could better showcase efficiency.

Overall the paper makes a reasonable contribution, but needs more work in showing the benefits of the approach as outlined above. A reject is recommended.

---

### Official Review · Reviewer_j83w · 2023-10-23
**Model compression by interactive knowledge distillation**

**Confidence:** 4

**Review:**

# Synopsis:

This paper introduces a three-step (all at once) framework for model compression in order to satisfy computing resource requirements. The setting comprises two subnetworks (learners) and a leader network (teacher). The two subnetworks are learning from the true model (self-learning) which happens along with communication between learners to enable performance balance by communicating soft labels (interactive learning) which also happens simultaneously with the leader learning from the true model (guided learning) due to insufficiency of balance accumulated in the interaction learning phase.

The motivation is well explained and very important, I believe it can be extended to many areas like annotating patient data in medical domains, where the models can be very complex (in terms of size) but still can do the job of producing a response based on each patient with a simpler compressed version of the complex model, hence allowing adaptivity depending on an available computing resource.

# Presentation (underlying some unclear points):

The authors did a nice job in presenting the paper in its hierarchical simple logic, however, many missing pieces (non-rigorous transition) can be observed from the paper, including explanations/ justifications for example: In equation (5), $s.r^{(j)}$, $r^{(j)}$  is the number of **repetitive** blocks at the j-th stage, the authors need to explain how the scaling factor is related to the number of repetitive blocks. In equation (9), the authors do not define $\pi_i$ and only state that it's some probability calculated by the mask.  In equation (13) the regularization term $\lambda$ was never used later for the approach, is there an explanation? The authors state "The proposed framework is not tied to any specific architecture but can incorporate any existing models/architectures", this is confusing when the problem isn't discussed in its general setting but in a setting that includes the concept class of a family of neural network architectures where talking about subnet makes sense having the scaling factor. There's a big gap between section 4.2 and section 4.3; from my understanding, the goal is to find the smallest subnet (in terms of computational complexity, width) that would accomplish the same mission as the leader but with less computations, it is not shown (understood) from the definition of your loss functions, it seems like you're training a bootstrap of subnets.
Also, it is not clear to me how the authors schedule interactions between learners/between learners and teachers.

In line 200, the authors state: " The same scaling factor is chosen to generate their sub-nets.", I'm wondering why you considered this choice, from my understanding this doesn't allow you to choose "the smallest possible circuit of your net", since the comparison is done based on a fixed scaling factor and doesn't allow comparison with subnets of smaller sizes.

Before introducing guided learning, the authors observed that a balance of losses may still not be preserved from the interaction learning phase, it would be interesting if you have a qualitative explanation of why that happens, given the empirical result in Table 2.

Despite the remarks, I think the work is interesting and well-motivated with an interesting approach but requires a rigorous setting for interactive learning.

---

### Official Review · Reviewer_waro · 2023-10-24
**Review submission 26**

**Confidence:** 1

**Review:**

This paper proposes a collaborative training method for training models of different sizes simultaneously. The framework involves utilizing three networks for knowledge exchange, wherein two of the networks are capable of creating subnetworks that can acquire knowledge from each other through knowledge distillation. The remaining network is responsible for learning from the true labels and guiding the learning process.

Strength :

The problem is motivated, and the paper is well-written. The technical aspects are clearly explained in a step-by-step manner, making it easy to understand the full setup. The study is presented well.

Comment :

Although I am a novice in this field and cannot provide an opinion on the novelty of this method, I would like to know more about the use of two teammate nets for exchanging knowledge. Considering that both nets were trained on the same dataset and may have the same capacity, I am unsure if distilling knowledge between them would be beneficial, as it would require additional training costs for the second model. Could the authors provide more information on this point? What would happen if the subnetworks learned from the logit of LeaderNet instead of the other teammate net?

---

### Official Review · Reviewer_J9HP · 2023-10-25
**The framework is general and provides more flexible choices for the model size, but the paper lacks clarity and  thoroughness.**

**Confidence:** 3

**Review:**

**Pros**:

The proposed cooperative learning framework is novel and interesting. It is independent of the model architecture and allows more flexible choices for the model size. Furthermore, the pipeline is a "package deal," producing an array of model choices spanning different sizes simultaneously.

**Cons**:

1. The authors need to enhance both the clarity of their writing and the overall presentation.

- Section 4 is unclear, making it hard for me to grasp their methodology fully. The global mask $B$ lacks a clear definition. The authors did not explain the meaning of $p$ in eq (9). Is it some learnable parameters of the mask $B$?  It seems that for a specified scaling factor, certain layers are *dynamically* activated within each block. Yet, Figure 2 (a) depicts all blocks activating only their initial layers. Such discrepancies between the text and the figures introduce confusion.
- In Section 5, labeling the conventional ResNet-152 training as the 'baseline' is misleading. Shouldn't methods like BranchyNet and BYOT also be considered baselines? The training process relying solely on interactive learning in Section 5.4.1 is perplexing. Given that equations (15) and (16) do not involve the actual label, how is the true label incorporated into the models without the leader net?

2. The empirical evaluation lacks thoroughness.

- The ablation studies cannot convince me of the necessity and effectiveness of all the components. For example, why do we need two teammate networks? If soft labels are preferred over hard ones, can we train only one teammate net and the leader net, with the latter's role being the generation of soft labels?
- The authors did not fully explain the experimental results. Specifically, why does COOP's accuracy initially rise and later decline with increasing parameters, while all three baselines show a consistent increase? Moreover, what leads to COOP consistently underperforming compared to BranchyNet across both datasets?  Though COOP does not have to beat all baselines considering its added benefits like generality and flexibility, the paper should still offer comprehensive explanations and a more detailed study.

---

### Meta-Review · Area_Chair_mMar · 2023-10-27

**Recommendation:** Accept (Poster)
**Confidence:** 4

**Metareview:**

The paper received mixed reviews, with Reviewer UwsV recommending a rejection. Reviewer J9HP described the work as "novel and interesting", Reviewer UwsV asses it as "appears novel" while R j83w as "motivation is well explained and very important". All Rs have some concerns with the empirical evaluation. Given that the method itself appears to have a novelty component, the empirical part is something the authors could improve post-acceptance, hence recommending for acceptance. The authors should carefully read the feedback provided and try to address it in the updated manuscript.

---

### Decision · Program_Chairs · 2023-10-28

**Decision:**

Accept (Poster)

**Comment:**

We thank the authors for their time and contribution to WANT and we are pleased to share that after the reviewing process the paper has been accepted. Congratulations! We encourage the authors to consider reviewers' feedback for the improvement of the camera-ready version. We hope to see you in person at the workshop and brainstorm on efficient training research together!